# Revisiting Active Sequential Prediction-Powered Mean Estimation

**Maria-Eleni Sfyraki**
University of California San Diego
msfyraki@ucsd.edu

**Jun-Kun Wang**
University of California San Diego
jkw005@ucsd.edu

## Abstract

In this work, we revisit the problem of active sequential prediction-powered mean estimation, where at each round one must decide the query probability of the ground-truth label upon observing the covariates of a sample. Furthermore, if the label is not queried, the prediction from a machine learning model is used instead. Prior work proposed an elegant scheme that determines the query probability by combining an uncertainty-based suggestion with a constant probability that encodes a soft constraint on the query probability. We explored different values of the mixing parameter and observed an intriguing empirical pattern: the smallest confidence width tends to occur when the weight on the constant probability is close to one, thereby reducing the influence of the uncertainty-based component. Motivated by this observation, we develop a non-asymptotic analysis of the estimator and establish a data-dependent bound on its confidence interval. Our analysis further suggests that when a no-regret learning approach is used to determine the query probability and control this bound, the query probability converges to the constraint of the max value of the query probability when it is chosen obliviously to the current covariates. We also conduct simulations that corroborate these theoretical findings.

## 1 Introduction

The mean estimation problem is a classical inference task that has seen revived interest in machine learning and statistics over the last few years. While the conventional setting is well-understood, numerous works have explored this problem under diverse settings and assumptions, aiming to enhance our understanding of the inherent challenges of learning from limited data. Recently, a line of work has investigated the design of efficient mean estimators under the robust framework, including the setting of a constant fraction of adversarial outliers (Cheng et al., 2020), heavy-tailed symmetric distributions without moment assumptions (Novikov et al., 2023), mean-shift contamination in multivariate identity Gaussian distributions (Diakonikolas et al., 2025), sparse mean estimation in high dimensions (Pensia, 2024), online high-dimensional mean estimation (Kane et al., 2024). A few other recent works have explored other structural considerations, such as collaborative normal mean estimation in the presence of strategic agents (Chen et al., 2023), communication-efficient mean estimation in a distributed setting (Ben-Basat et al., 2024), vector mean estimation under the shuffle model of privacy (Asi et al., 2024), dynamic multi-group mean estimation (Aznag et al., 2023), leveraging favorable distribution structure to improve sub-Gaussian rate (Dang et al., 2023), among others. This variety of scopes in the mean estimation setup highlights its relevance as a foundational task for inference.

A direction that has attracted substantial attention with the increasing integration of machine learning is mean estimation through an active inference perspective (Zrnic & Candes, 2024). Specifically, the problem focuses on estimating the mean label from a set of unlabeled observations, by leveraging a limited label collection budget and the abundant but potentially biased predictions of a machine learning model. Under this setup, active statistical inference provides a data collection strategy that utilizes the budget more effectively by taking into consideration which labels it would be more beneficial to acquire. Specifically, it prioritizes the collection of labels where the model exhibits higher uncertainty and uses the model predictions for instances where the model is more confident. The same work considers querying the labels in both the batch and sequential setting, where the latter

additionally allows updating the model as the ground-truth labels are obtained, and provides *asymptotically* valid confidence intervals for the estimator in question in both cases. The non-asymptotic analysis of the estimator was not provided in the prior work. [1]

In this work, we draw on the sequential active statistical inference perspective by providing *non-asymptotic* guarantees for the sequential active mean estimation problem. While prior work of Zrnic & Candes (2024) have established the asymptotic normality of their proposed estimator, our work investigates the sequential mean estimation problem further under the light of an online updating scheme and provides a non-asymptotic analysis with guarantees that hold *at any time* while going through the data. More specifically, we first formulate the scheme of active sequential mean estimation as an online update step, and establish a convergence guarantee that incorporates the conditional variance of the update direction and achieves a rate of $\tilde{O}\left(1/\sqrt{t}\right)$ for sufficiently large $t$.

Furthermore, motivated by a series of experimental findings, which reveal an intriguing pattern of the label sampling rule considered by previous work, we are led to examine more closely the role of the model uncertainty component through the current covariate. In particular, to adhere to the budget constraint and ensure a small variance of the estimator, Zrnic & Candes (2024) derive a sampling rule that is a mixture of the uniform rule and a model uncertainty estimator weighted by a mixing constant. Our experimental findings across a variety of tasks indicates that employing the mixture rule or relying solely on the uniform policy results in comparable confidence interval widths, with the uniform policy occasionally producing marginally narrower intervals. This observation suggests that, the contribution of model uncertainty with respect to the label of the current covariate might be brittle in practice. Based on this insight, we formulate the problem of tuning the query policy as an online learning task that does not rely on the current covariate, and whose validity is supported by the strong sublinear regret guarantees of the classical Follow-the-Regularized-Leader (FTRL) algorithm (Abernethy et al., 2012). Remarkably, we demonstrate that under this no-regret learning approach, the query policy converges to the maximum value permitted by the budget constraint. Our theoretical findings are further validated through experiments on three real-world and one synthetic dataset.

## 2 PRELIMINARIES AND NOTATION

We begin by introducing the problem setup. We consider the setting where we have access to a sequence of data points $x_1, x_2, \ldots x_T \in \mathsf{X} \subset \mathbb{R}$ from an unknown, fixed distribution $\mathbb{P}_X$. Each data point $x_t$ is associated with a ground-truth label $y_t \in \mathsf{Y} \subset \mathbb{R}$, drawn from an also unknown, fixed distribution $\mathbb{P}_{Y|X}$. We assume that the ground-truth labels are not known a priori, and the cost to obtain them could be high. We are interested in estimating the mean label $\mu_y = \mathbb{E}[y_t]$. Additionally, we assume that at each time $t \in [T]$ we have access to a black-box predictive model $f_t(\cdot) : \mathsf{X} \to \mathsf{Y}$, which can continually evolve by using the samples collected up to round $t-1$ to update. More specifically, we require $f_t \in \mathcal{F}_{t-1}$, where $\mathcal{F}_t$ denotes the $\sigma$-algebra generated by the first $t$ data points $x_s, 1 \le s \le t$.

Sequential active mean estimation seeks to construct an efficient estimator by observing data points one at a time and deciding whether to query each ground-truth label. Under a limited labeling budget, the objective is to sequentially acquire labels in a way that most effectively improves the accuracy of the mean estimator. Specifically, if we denote by $T_{lab}$ the total number of collected labels, we require that the policy of querying the ground-truth label ensures $\mathbb{E}[T_{lab}] \le T_b$, where we assume that $T_b \ll T$. Let $\pi_t(x_t)$ denote the probability of collecting the label of data point $x_t$ at time $t$, where $\pi_t \in \mathcal{F}_{t-1}$, and let $\xi_t \sim \text{Bernoulli}(\pi_t(x_t))$ denote the labeling decision used to indicate whether the ground-truth label $y_t$ was collected ($\xi_t = 1$) or not ($\xi_t = 0$). Zrnic & Candes (2024) propose the sequential active mean estimator $\hat{w} = \frac{1}{T} \sum_{t=1}^{T} \left( f_t(x_t) + (y_t - f_t(x_t)) \frac{\xi_t}{\pi_t(x_t)} \right)$. It is easy to verify that $\hat{w}$ is unbiased, i.e., $\mathbb{E}[\hat{w}] = \mu_y$. Notably, Zrnic & Candes (2024) show that the optimal choice of querying policy $\pi_t^{opt}$ is according to the uncertainty of the prediction model, and

---

[1] However, we acknowledge that Appendix C of Zrnic & Candes (2024) outlines a related scheme based on estimating bounded means via testing by betting (Waudby-Smith & Ramdas, 2024), which may come with certain non-asymptotic guarantees. We further provide a discussion in Appendix B.

it satisfies

$$\pi_t^{opt}(x_t) \propto \sqrt{\mathbb{E}\left[(y_t - f_t(x_t))^2 \,|\, \mathcal{F}_{t-1}\right]},$$

where the above expression hides a normalization constant to ensure that $\mathbb{E}[\pi_t^{opt}(x_t)] \leq T_b/T$. However, since $\mathbb{P}_{Y|X}$ is unknown, the authors suggest fitting a model on past data $(x, y)$ to approximate the uncertainty $u_t(x_t)$ by $|y_t - f_t(x_t)|$ for a given $x_t$, and then setting the querying policy to be proportional to that estimate. To ensure that the budget constraint is met in practice, the remaining budget at time $t$ is set as the difference between the expected budget to be used up to time $t$ and the budget already used up to time $t - 1$, i.e. $T_{\Delta,t} = tT_b/T - T_{lab,t-1}$. Then, the querying policy is set to

$$\pi_t(x_t) = \min\left\{\eta_t u_t(x_t), T_{\Delta,t}\right\}_{[0,1]},$$

where the subscript $[0, 1]$ denotes clipping to $[0, 1]$, $\eta_t$ is a normalizing constant set to $\eta_t = T_b/(T\mathbb{E}[u_t(x_t)])$, and $\mathbb{E}[u_t(x_t)]$ is approximated empirically. This practical rule aims to balance frequent sampling under high uncertainty against overusing the budget. However, since the estimated uncertainty can be consistently low, and thus the budget could be underutilized, the policy is occasionally set to $\pi_t(x_t) = (T_{\Delta,t})_{[0,1]}$. An additional concern is that an inaccurate uncertainty estimate may be close to zero, while the quantity $|y_t - f_t(x_t)|$ is actually large, which would in turn yield an amplified estimator variance. To address this issue, the authors suggest that the policy is set to a mix of the described policy with the uniform rule as

$$\pi_t^{(\lambda)}(x_t) = (1 - \lambda)\pi_t(x_t) + \lambda\pi_t^{\text{unif}}(x_t),$$

where $\pi_t^{\text{unif}}(x_t) = T_b/T$, and $\lambda \in [0, 1]$. In the presence of sufficient historical data, $\lambda$ can be tuned by minimizing the empirical estimate of the estimator variance induced by the policy $\pi_t^{(\lambda)}$. However, due to insufficient historical data in the sequential setting, $\lambda$ is set to a fixed value.

While the related work simply sets the mixing parameter to $0.5$, we explore different parameter values for the mean estimation experiment in Zrnic & Candes (2024) by running their public implementation on the same post-election survey dataset Center (2020) as in their experiments. We keep all their other parameter choices unchanged. Figure 1 shows how the interval width varies with the sampling budget $T_b$ under different values of the mixing parameter in the query policy [2]. We find that setting $\lambda = 1$, corresponding to using the uniform query policy that ignores model uncertainty, produces confidence intervals that are slightly narrower than those obtained with $\lambda = 0.5$. We further evaluate the method on two additional real-world datasets and one synthetic dataset, and observe a similar pattern, where the uniform query policy ($\lambda = 1$) yields confidence intervals that are comparable to, and often tighter than those from $\lambda = 0.5$. Due to space constraints, the corresponding figures are included in Appendix A.

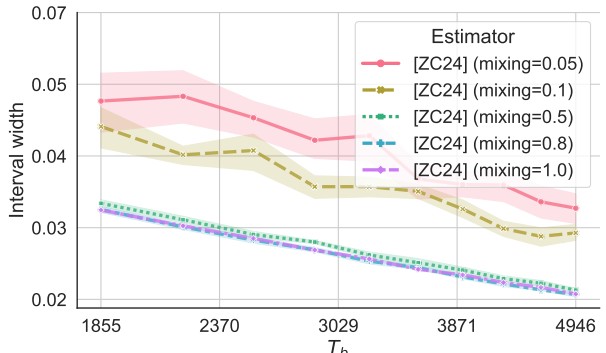

Figure 1: **Post-election survey dataset.** Interval width vs. the sampling budget parameter $T_b$ for different values of the mixing parameter of the query probability scheme in Zrnic & Candes (2024). Averaged over 10 repeated runs.

These similar empirical patterns motivate us to further investigate this observation. A plausible initial explanation is that the empirical performance improves when less weight is assigned to the uncertainty-based component, possibly because the uncertainty predictor may not be reliable. However, our theoretical analysis later suggests that this factor alone may not account for the phenomenon, which might be surprising.

---

[2]The code, provided in Jupyter notebook format, to reproduce Figure 1 as well as others in the paper is available in the supplementary material.

## 3 RELATED WORK

**Active Statistical Inference.** The ideas in this work are motivated by the recent approach of Active Statistical Inference (Zrnic & Candes, 2024). Extending this framework, Angelopoulos et al. (2025) propose a method that optimizes the sampling rate between gold-standard and pseudo-labels rather than relying on a fixed label budget, and derive an improved active sampling policy. A recent work by Gligorić et al. (2025) utilizes LLM verbalized confidence scores to guide their sampling policy and subsequently performs active inference by combining the LLM and human annotations.

**Prediction Powered Inference.** The Active Statistical Inference framework is grounded in the idea of Prediction Powered Inference (PPI) Angelopoulos et al. (2023a), which differs from the former in that it assumes the availability of a small, pre-labeled dataset. The work of Angelopoulos et al. (2023b) introduces PPI++, which improves the computational efficiency of PPI by addressing the intractability of the original confidence interval construction and using a tuning parameter to control the influence of the model predictions based on their quality. Subsequent works (Dorner et al., 2025; Mani et al., 2025) analyze the critical role of the correlation between the gold-standard and model-generated labels for the performance of PPI. Following the original PPI formulation, several works have proposed extensions and refinements in various directions, including addressing estimator bias in the few-label regime (Eyre & Madras, 2025), incorporating an inverse probability weighting (IPW) bias-correction term (Datta & Polson, 2025), combining predictions from multiple foundation models via a hybrid augmented IPW estimator (De Bartolomeis et al., 2025), applying a stratification approach (Fisch et al., 2024), exploring a bootstrap-based variant (Zrnic, 2024), employing Bayes-assisted approaches (Cortinovis & Caron, 2025; Li & Ignatiadis, 2025) and extending the ideas of PPI to e-values (Csillag et al., 2025). Other applications of the PPI framework include LLM-assisted rank-set construction (Chatzi et al., 2024), average treatment effects from multiple datasets (Wang et al., 2025), clinic trial outcomes (Poulet et al., 2025), autoevaluation in machine learning systems (Boyeau et al., 2025; Park et al., 2025), machine learning generated surrogate rewards for multi-armed bandits (Ji et al., 2025b). A few other works have explored alternative machine-learning assisted estimators, e.g., Schmutz et al. (2023); Egami et al. (2023); Miao et al. (2023); Miao & Lu (2024); Gan et al. (2024).

We refer the reader to Appendix B for a more detailed discussion on related work.

## 4 NON-ASYMPTOTIC ANALYSIS OF THE MEAN ESTIMATOR

In this section, we provide the non-asymptotic analysis of the sequential active mean estimator. Since the sequential active estimation setting requires going over the data points sequentially, we can formulate the active mean estimator of Zrnic & Candes (2024) as an online update step at each time $t \in [T]$, as

$$w_{t+1} = w_t + \frac{1}{T}\left(f_t(x_t) + (y_t - f_t(x_t))\frac{\xi_t}{\pi_t(x_t)}\right), \tag{1}$$

where we set the initial point $w_1 = 0$, and let $T$ be the horizon. We denote $g_t := f_t(x_t) + (y_t - f_t(x_t))\frac{\xi_t}{\pi_t(x_t)}$. A simple calculation shows that $\mathbb{E}[g_t] = \mu_y$, the mean of the random variable $(y_t)_{t \geq 1}$.

Before proceeding with the asymptotic analysis of (1), we will need a technical lemma, known as Freedman's inequality, which is stated below for completeness.

**Lemma 1** (Freedman's inequality (Freedman, 1975), see also e.g., Lemma 3 in Rakhlin et al. (2012)). *Let $\zeta_1, \ldots, \zeta_T$ be a martingale difference sequence with a uniform upper bound $|\zeta_t| \leq b, \forall t$. Denote $V_t$ the sum of conditional variances of $\zeta_s$'s., i.e., $V_t = \sum_{s=1}^{t} \text{Var}(\zeta_s|\zeta_1, \ldots, \zeta_{s-1})$. Also, denote $\sigma_t := \sqrt{V_t}$. Then, for any $0 < \delta < 1/e$ and $T \geq 4$, we have*

$$\text{Prob}\left(\exists t \leq T : \sum_{s=1}^{t} \zeta_s \geq 2\max\left\{2\sigma_t, b\sqrt{\log(1/\delta)}\right\}\sqrt{\log(1/\delta)}\right) \leq \log(T)\delta.$$

Lemma 1 provides a concentration inequality for martingales, yielding a high-probability bound on the deviation of a martingale sum from its mean that adapts to the accumulated conditional

variance. We are now ready to present the non-asymptotic analysis result for the sequential active mean estimator, as detailed in Theorem 1

**Theorem 1.** *Fix a time horizon $T \geq 4$. Assume each random variable $g_t$ is bounded, i.e., $|g_t| \leq G$ for a constant $G > 0$. Denote $\sigma_t^2 := \mathbb{E}\left[(g_t - \mu_y)^2 | \mathcal{F}_{t-1}\right]$ the conditional variance, where $\mathcal{F}_{t-1}$ is the filtration up to $t-1$. Then, for any $\delta \in (0, 1/e)$, with probability at least $1 - \delta$, $\forall t \in [T]$ :*

$$|w_{t+1} - \mu_y| \leq \frac{2 \max\left\{2\sqrt{S_t}, (G + |\mu_y|)\sqrt{\log\left(\frac{\log(T)}{\delta}\right)}\right\} \sqrt{\log\left(\frac{\log(T)}{\delta}\right)}}{T} + \left(1 - \frac{t}{T}\right)|\mu_y|, \tag{2}$$

*where $S_t = \sum_{s=1}^{t} \sigma_s^2$.*

Theorem 1 demonstrates a data-dependent bound on the accuracy of the update $w_t$ that holds with high-probability at any time $t \in [T]$. We make a couple remarks on this result. When $t \ll T$ and the $\left(1 - \frac{t}{T}\right)|\mu_y|$ term dominates the other term on (2), it is possible to observe a rate that is slower than $O(1/\sqrt{t})$ in the initial stage, i.e., $1 - \frac{t}{T} \approx 1$. However, after this burn-in stage (i.e., when $t$ is sufficiently large), the first term will eventually become dominant. Furthermore, when $\sqrt{\sum_{s=1}^{t} \sigma_s^2}$ dominates $(G + |\mu_y|)\sqrt{\log(\log(T)/\delta)}$, which happens easily when $t$ is sufficiently large and $\delta$ is not too small, the rate becomes

$$|w_{t+1} - \mu_y| = O\left(\frac{\sqrt{\sum_{s=1}^{t} \sigma_s^2}\sqrt{\log(\log(T)/\delta)}}{T}\right). \tag{3}$$

Using the trivial bound $\sum_{s=1}^{t} \sigma_s^2 \leq 2t(G^2 + \mu_y^2)$, we can further express the rate as $O\left(\frac{\sqrt{t}\sqrt{(G^2 + \mu_y^2)\log(\log(T)/\delta)}}{T}\right) = O\left(\frac{1}{\sqrt{t}}\right)$.

## 5 POLICY OF QUERYING THE GROUND TRUTH

In the previous section, we discussed that the update $w_t$ of (1) will have a rate of $O\left(\sqrt{t}/T\right) = O\left(1/\sqrt{t}\right)$ in the worst case. An observation is that when $\sum_{s=1}^{t} \sigma_s^2 \ll 2t(G^2 + \mu_y^2)$, one might get an even faster rate than $O(1/\sqrt{t})$. This motivates us to control the sum of the conditional variances, i.e., $\sum_{s=1}^{t} \sigma_s^2 = \sum_{s=1}^{t} \mathbb{E}\left[(g_s - \mu_y)^2 | \mathcal{F}_{s-1}\right]$, by proposing an algorithm to determine the query probability of the ground-truth label online, which we detail next.

We begin by introducing the following observation on the decomposition of the conditional variance of the update step, which will subsequently guide the choice of the online query policy.

**Lemma 2.** *The conditional variance has the following decomposition:*

$$\mathbb{E}\left[(g_t - \mu_y)^2 | \mathcal{F}_{t-1}\right] = \mathbb{E}\left[f_t(x_t)^2 \Big| \mathcal{F}_{t-1}\right] + \mathbb{E}\left[(y_t - f_t(x_t))^2 \frac{1}{\pi_t(x_t)} \Big| \mathcal{F}_{t-1}\right]$$

$$+ 2\mathbb{E}\left[f_t(x_t)(y_t - f_t(x_t)) \Big| \mathcal{F}_{t-1}\right] - \mu_y^2.$$

*Furthermore, assume that the query policy at time $t$ is $\mathcal{F}_{t-1}$-measurable, i.e., there exists a random variable $p_t \in [0, 1]$, measurable with respect to $\mathcal{F}_{t-1}$, such that $\pi_t(x_t) = p_t$. Then, we have*

$$\mathbb{E}\left[(y_t - f_t(x_t))^2 \frac{1}{\pi_t(x_t)} \Big| \mathcal{F}_{t-1}\right] = \frac{1}{p_t}\mathbb{E}\left[(y_t - f_t(x_t))^2 \Big| \mathcal{F}_{t-1}\right].$$

We observe that the only term involving the query probability $\pi_t(x_t)$ that contributes to the conditional variance is $\mathbb{E}[(y_t - f_t(x_t))^2 \frac{1}{\pi_t(x_t)}|\mathcal{F}_{t-1}]$. We now consider the query policy at time $t$ that is fully determined by the information up to $t-1$. With this, we can rewrite $\mathbb{E}[(y_t - f_t(x_t))^2 \frac{1}{\pi_t(x_t)}|\mathcal{F}_{t-1}] = \frac{1}{p_t}\mathbb{E}[(y_t - f_t(x_t))^2 |\mathcal{F}_{t-1}]$. However, we note that $\mathbb{E}[(y_t - f_t(x_t))^2 |\mathcal{F}_{t-1}]$ cannot be known since this depends on the unknown distributions of $y_t$ and $f_t(x_t)$. Therefore, we assume that there is an oracle, denoted as $\Phi_t(x_t) \in \mathbb{R}_+$, which is available at time $t$ and provides an approximation of the quantity of interest, i.e.,

$$\frac{1}{c_1}\Phi_t(x_t) \leq \mathbb{E}\left[(y_t - f_t(x_t))^2 \,\middle|\, \mathcal{F}_{t-1}\right] \leq c_0\Phi_t(x_t),$$

for some constants $c_0, c_1 > 0$ such that $\Phi_t(x_t) \approx \mathbb{E}[(y_t - f_t(x_t))^2 |\mathcal{F}_{t-1}]$. Equipped with such an oracle, we propose specifying the query probability $p_t$ based on the following rule:

$$p_t \leftarrow \arg \min_{p \in [\beta, \tau]} \gamma\theta_{t-1}p + \frac{1}{2}p^2, \quad \text{where } \theta_{t-1} := -\sum_{s=1}^{t-1} \frac{\Phi_s(x_s)}{p_s^2}, \tag{4}$$

where $\gamma > 0$, $\tau \in (0, 1]$, $\beta \in (0, \tau]$ are user-specified parameters, and we let $\theta_0 := 0$. The following lemma shows that $p_t$ has a closed-form expression.

**Lemma 3.** *The update (4) has a closed-form expression, which is*

$$p_t = \max\{\beta, \min\{\tau, -\gamma\theta_{t-1}\}\}.$$

We note that one can specify $\tau = \frac{T_b}{T}$, where $T_b$ denotes the targeted maximum number of rounds in which the ground-truth is queried, which ensures that the query probability $p_t$ at time $t$ does not exceed the ratio $\frac{T_b}{T}$. This constraint is also akin to the sampling rule $\mathbb{E}[p_t] \leq \frac{T_b}{T}$ considered in Zrnic & Candes (2024). On the other hand, the parameter $\beta$ encourages certain exploration at each round by preventing the query probability $p_t$ from becoming too close to 0.

The update (4), in a nutshell, is one of the celebrated online learning algorithms called Follow-the-Regularized-Leader (FTRL), (see, e.g., Abernethy et al. (2012); Wang et al. (2024), and Chapter 7 of Orabona (2019)). FTRL is known to enjoy a sublinear regret bound when the sequence of loss functions is convex. We propose leveraging the strong guarantee of FTRL to determine the query probability $p_t$ online. More specifically, in our scenario, one first determines the query probability $p_t$, after which it receives a loss function defined as $\tilde{\ell}_t(p) := \frac{\Phi_t(x_t)}{p}$, which is a convex loss function in $(0, 1]$. In online learning, a common goal is to minimize the regret. In our setting, the regret against a benchmark $p_* \in [\beta, \tau]$ over $t$ rounds is defined as:

$$\text{Regret}_t(p_*) := \sum_{s=1}^{t} \tilde{\ell}_s(p_s) - \sum_{s=1}^{t} \tilde{\ell}_s(p_*) = \sum_{s=1}^{t} \frac{\Phi_s(x_s)}{p_s} - \sum_{s=1}^{t} \frac{\Phi_s(x_s)}{p_*}, \tag{5}$$

where the first sum is the cumulative loss of the updates $(p_s)_{s \geq 1}$ and the second one is that of the benchmark. A sublinear regret bound against any benchmark $p_*$ in the same decision space $[\beta, \tau]$ of the learner implies that the sequence of query probabilities can *compete* with the best fixed query probability in hindsight. On the other hand, given that the oracle's output is non-negative, i.e., $\forall s : \Phi_s(\cdot) \geq 0$, it follows that $\arg \min_{p_* \in [\beta, \tau]} \tilde{\ell}_s(p) = \tau$. Combining these implies that an online learner may need to approach $\tau$ eventually to achieve a sublinear regret. In other words, to maintain sublinear regret, the query probabilities $p_t$ will need to converge toward the constraint upper bound $\tau = \frac{T_b}{T}$. In particular, we have that the average regret is in fact vanishing (a.k.a. no-regret learning), i.e., $\frac{\text{Regret}_t(p_*)}{t} \to 0$ as $t \to \infty$, as the following lemma shows.

**Lemma 4.** *(see e.g., Theorem 3 in Luo (2017)) FTRL satisfies*

$$\text{Regret}_t(p_*) \leq \gamma \sum_{s=1}^{t} \left|\dot{\tilde{\ell}}_s\right|^2 + \frac{R(p_*) - \min_{p \in \mathcal{K}} R(p)}{\gamma},$$

*for any comparator $p_* \in \mathcal{K} := [\beta, \tau]$, where $\gamma > 0$, $R(p) := \frac{1}{2}p^2$, and $\dot{\tilde{\ell}}_s := \frac{d\tilde{\ell}_s(p)}{dp}\Big|_{p=p_s}$.*

We note that Lemma 4 is a classical result in online learning literature, see also Orabona (2019); Shalev-Shwartz et al. (2012). The guarantee suggests that if one chooses $\gamma = \frac{1}{\sqrt{T}}$, then the regret of FTRL is $O(\sqrt{T})$, which grows sublinearly with $T$, provided that the size of the derivative is bounded. The following lemma shows that the size of the derivative in the regret bound is bounded whenever the oracle's output is bounded.

**Lemma 5.** *Assume that the range of oracle's output is bounded, i.e., $\forall t : \Phi_t(x_t) \leq B$, for a constant $B > 0$. Then, $\forall t : \left|\dot{\tilde{\ell}}_t\right|^2 \leq \frac{B^2}{\beta^4}$.*

In the following theorem, we denote $\sigma_{1:t}^{*2} := \sum_{s=1}^{t} \sigma_{s,(t)}^{*2}$, the cumulative conditional variance obtained under a fixed query probability $p_{1:t}^* \in \mathcal{K} := [\beta, \tau]$, as if the method had committed the best fixed probability *in hindsight* over $t$ rounds rather than following the query policy from (4), i.e., $p_{1:t}^* = \arg\min_{p \in [\beta, \tau]} \sum_{s=1}^{t} \tilde{\ell}_s(p)$.

**Theorem 2.** *Assume that there is an oracle that outputs $\Phi_t(x_t)$ at each $t$ such that $\frac{1}{c_1}\Phi_t(x_t) \leq$*

$$\mathbb{E}\left[(y_t - f_t(x_t))^2 \mid \mathcal{F}_{t-1}\right] \leq c_0 \Phi_t(x_t) \text{ for some constant } c_0, c_1 > 0 \text{ and that } \forall t : \Phi_t(x_t) \leq B.$$

*Set the parameter $\gamma = \frac{1}{\sqrt{T}}\frac{\beta^2}{B}$. Using the query policy (4), we have that, with probability at least $1 - \delta$, $\forall t \in [T]$:*

$$|w_{t+1} - \mu_y| \leq \frac{2\max\left\{2\sqrt{\Psi_t}, (G + |\mu_y|)\sqrt{\log\left(\frac{\log(T)}{\delta}\right)}\right\}\sqrt{\log\left(\frac{\log(T)}{\delta}\right)}}{T} + \left(1 - \frac{t}{T}\right)|\mu_y|,$$

*for any $\beta \in (0, \tau)$ and $\tau \in (0, 1)$, where $\Psi_t \leq c_0 c_1 \sigma_{1:t}^{*2} + 2c_0 \frac{t}{\sqrt{T}}\frac{B}{\beta^2} + \frac{c_0(R(p_{1:t}^*) - \min_{p \in \mathcal{K}} R(p))\sqrt{T}B}{\beta^2}$.*

What Theorem 2 shows is a *data-dependent* bound. We note that $\sqrt{\Psi_t} \leq \sqrt{c_0 c_1 \sigma_{1:t}^{*2}} + O\left(T^{1/4}\right)$. From our earlier discussion, once a sufficient burn-in period has elapsed so that the first term in the upper bound dominates, the non-asymptotic rate takes the form $O\left(\frac{\sqrt{\Psi_t}}{T}\right) = O\left(\frac{\sqrt{c_0 c_1 \sigma_{1:t}^{*2}}}{T} + \frac{1}{T^{3/4}}\right)$, provided that $\delta$ is not too small.

## 6 EXPERIMENTS

In this section, we report experimental results by comparing the proposed method with two baselines. For clarity, Algorithm 1 presents the protocol for the task of active sequential mean estimation. Compared to the procedure described in Algorithm 2 of Zrnic & Candes (2024), the key difference is that we also update the uncertainty predictor whenever the ML model for label prediction is updated. Furthermore, we split the dataset with ground-truth labels into two disjoint subsets, which are accumulated as described on Line 13, and use these subsets to expand the data available for updating the ML model $f_{t+1}$ and its uncertainty predictor $u_{t+1}$, respectively. This treatment of the disjoint training sets is intended to enable the uncertainty predictor to more accurately estimate the uncertainty of the ML model when it is applied to *unseen* data at test time.

The first baseline was also considered in the prior work of Zrnic & Candes (2024).

$$w_T^{\text{Uniform}} := \frac{1}{T}\sum_{t=1}^{T}\left(f(x_t) + \frac{(y_t - f(x_t))\xi_t}{T_b/T}\right), \quad \text{where } \xi_t \sim \text{Bernoulli}\left(\frac{T_b}{T}\right) \tag{6}$$

For this baseline, we note that the ML predictor is fixed. As discussed in Zrnic & Candes (2024), this comparison can showcase the benefit of data collection. Following the terminology of Zrnic & Candes (2024), we refer to this baseline as "uniform sampling."

---

**Algorithm 1** Protocol of Active Sequential Mean Estimation

---

**Require:** Significance level parameter $\alpha \in (0, 1)$, target sampling budget $T_b > 0$, and batch size $B$.

1: **Initialize** a machine learning (ML) model $f_1(\cdot) : \mathsf{X} \to \mathsf{Y}$ to predict the labels of data.
2: **Initialize** an uncertainty predictor $u_1(\cdot, \cdot) : \mathsf{X} \times f_1(\cdot) \to \mathbb{R}_+$ for the model's predictions.
3: **Initialize** the dataset for updating the model $\mathcal{D}_{\text{train}}$ and the dataset for the uncertainty predictor $\mathcal{D}_{\text{uncertainty}}$.
4: **Set** $D_{\text{tmp}} \leftarrow \emptyset$, $w_1 \leftarrow 0$
5: **for** $t = 1, \ldots, T$ **do**
6:     Observe features $x_t$ of a sample and determine the query probability $p_t$ for getting its label.
7:     Sample the binary random variable $\xi_t \sim \text{Bernoulli}(p_t)$.
8:     **if** $\xi_t = 1$ **then**
9:         Obtain the ground-truth label $y_t$ and set $\mathcal{D}_{\text{tmp}} \leftarrow \mathcal{D}_{\text{tmp}} \cup \{(x_t, y_t)\}$.
10:        Increase $b$ by 1.
11:     **end if**
12:     **if** $B = b$ **then**
13:         Randomly split $\mathcal{D}_{\text{tmp}}$ into two datasets with equal sizes, $\mathcal{D}_{\clubsuit}$ and $\mathcal{D}_{\spadesuit}$.
14:         Set $\mathcal{D}_{\text{train}} \leftarrow \mathcal{D}_{\text{train}} \cup \mathcal{D}_{\clubsuit}$ and set $\mathcal{D}_{\text{uncertainty}} \leftarrow \mathcal{D}_{\text{uncertainty}} \cup \mathcal{D}_{\spadesuit}$.
15:         Update the ML model to $f_{t+1}(\cdot)$ using the dataset $\mathcal{D}_{\text{train}}$; similarly, update the uncertainty predictor $u_{t+1}(\cdot, \cdot) : \mathsf{X} \times f_{t+1}(\cdot) \to \mathbb{R}_+$ using the dataset $\mathcal{D}_{\text{uncertainty}}$.
16:         Reset $b \leftarrow 0$.
17:     **else**
18:         $f_{t+1} \leftarrow f_t$ and $u_{t+1} \leftarrow u_t$.
19:     **end if**
20:     Update the estimate $w_{t+1} = w_t + \frac{1}{T}\left(f_t(x_t) + (y_t - f_t(x_t))\frac{\xi_t}{p_t}\right)$.
21: **end for**
22: **Set** $\hat{\sigma}^2 \leftarrow \frac{1}{T}\sum_{t=1}^{T}\left(f_t(x_t) + (y_t - f_t(x_t))\frac{\xi_t}{p_t} - w_{T+1}\right)^2$.
23: **Output:** $(1 - \alpha)$-confidence interval $CI_\alpha = \left(w_{T+1} \pm z_{1-\alpha/2}\frac{\hat{\sigma}}{\sqrt{T}}\right)$.

---

The second baseline is the scheme proposed in Zrnic & Candes (2024) for implementing Line 6 in Algorithm 1, which determines the query probability $p_t$, as described in the earlier preliminary section.

### 6.1 DATASETS AND EXPERIMENTAL SETUP

We compare the algorithms on three real-world datasets. The first dataset concerns the politeness scores of texts based on human annotations, which is available from the works of Ji et al. (2025a) and Gligorić et al. (2025). For each article, there is an associated 21-dimensional feature vector and a score predicted by ChatGPT. We consider the task of regression for this dataset, where the ML model is trained on the 22-dimensional vector (the 21 features plus the ChatGPT score) to predict the average score of 5 human judgments. Following the suggestion in Zrnic & Candes (2024), an uncertainty estimator $u_t(\cdot, \cdot) : \mathsf{X} \times f_t(\cdot) \to \mathbb{R}_+$ is used to predict the absolute error $|f_t(x_t) - y_t|$ from $x_t$ without seeing the label $y_t$ beforehand. This predicted uncertainty is then used as the input to their proposed scheme for determining the query probability $p_t$ at round $t$. The uncertainty estimator is also updated based on the collected samples with queried ground-truth labels once every batch of $B$ labeled samples is collected, as depicted in Algorithm 1. For our proposed scheme, we need to construct the approximation oracle $\Phi_t(x_t)$. In practice, we implement this by performing a linear regression on the squared residual error $(f_t(x_t) - y_t)^2$ for the samples in $\mathcal{D}_{\text{uncertainty}}$, and this estimator is updated regularly after every batch of size $B$.

The second dataset concerns predicting the ratings of wine reviews, which is available in Ji et al. (2025a). Each review is associated with the price of the wine and four additional binary attributes representing the regions, along with the rating predicted by OpenAI's GPT-4o mini based on the reviewers' comments. We also consider the task of regression for this dataset, where a linear regression model is trained on the aforementioned covariates to predict the human ratings. The uncertainty

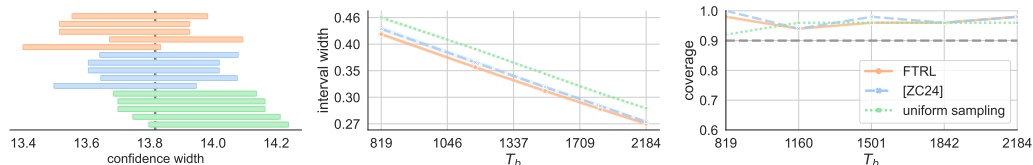

Figure 2: **Politeness score analysis**. Left: Intervals of randomly selected trials. Middle: Average confidence width across repeated trials vs. sampling budget $T_b$. Right: Percentage of trials that cover the true mean.

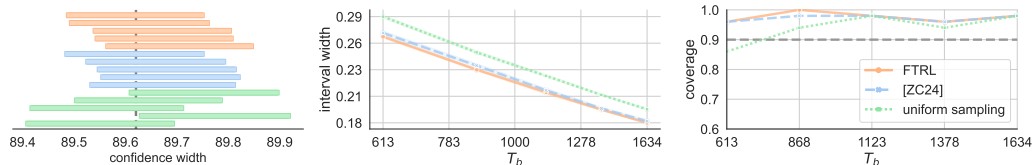

Figure 3: **Wine review analysis**. Left: Intervals of randomly selected trials. Middle: Average confidence width across repeated trials vs. sampling budget $T_b$. Right: Percentage of trials that cover the true mean.

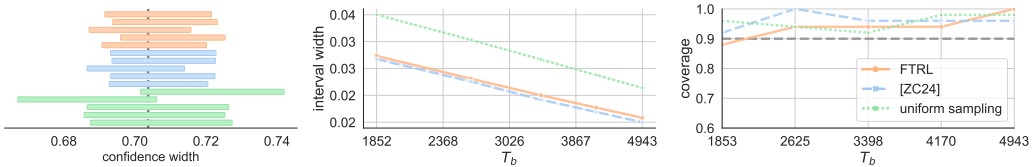

Figure 4: **Post-election survey.**. Left: Intervals of randomly selected trials. Middle: Average confidence width across repeated trials vs. sampling budget $T_b$. Right: Percentage of trials that cover the true mean.

predictor and the approximation oracle $\Phi_t(x_t)$ have the same form and are updated in the same fashion as for the first dataset.

The third dataset is a post-election survey dataset considered in Zrnic & Candes (2024), where the original source is from Center (2020). This dataset includes the approval ratings of two politicians, where approval is represented by $y_t \in \{0, 1\}$. Following the experimental setup in Zrnic & Candes (2024), the ML model $f(\cdot)$ is implemented as an XGBoost model. Since the response $y_t$ is binary, the task can be treated as a classification problem. We hence follow the treatment in Zrnic & Candes (2024) by using the uncertainty predictor as $u_t(x_t, f_t(\cdot)) = 2 \min\{f_t(x_t), 1 - f_t(x_t)\}$ for their proposed scheme, where $f_t(x_t)$ is the predicted probability of $y_t = 1$ for $x_t$ given by the XGBoost model. On the other hand, the required approximation oracle $\Phi_t(x_t)$ is trained in the same fashion as in the first two tasks.

We also conduct experiments on a synthetic dataset, the details of which can be found in Appendix E.

## 6.2 RESULTS

In this subsection, we report the results of the conducted experiments. Figures 2 - 4 and Figure 6 (provided in Appendix E due to space limitations) show the intervals of randomly selected trials, average confidence width, and coverage for each of the datasets considered over 50 trials. Across all four datasets examined, we find that the FTRL policy yields performance comparable to the mixture policy proposed by Zrnic & Candes (2024), in the sense that both result in confidence intervals of similar width, while both outperform the baseline policy. Notably, in two of the datasets, the FTRL policy attains marginally narrower confidence intervals. With respect to coverage of the true mean, all three policies yield a high proportion of confidence intervals that successfully include the true value.

Our theoretical analysis and experimental findings consistently indicate that when the query probability $p_t$ at time $t$ is oblivious to the current covariates $x_t$, while still permitted to depend on past covariates or past uncertainty estimates, the optimal strategy is simply to set $p_t = \frac{T_b}{T}$ in accordance with the sampling budget. This result, implies that constructing an uncertainty predictor, or leveraging uncertainty estimates in any form, does not appear to provide a clear advantage for this type

of policy. Perhaps unexpectedly, this rules out any benefit from conditioning on past covariates or past uncertainty estimates. Furthermore, as our figures illustrate, even when the query probability ignores the current covariates, FTRL, which quickly converges to the constant $\frac{T_b}{T}$ and then maintains it, performs on par with the more sophisticated scheme of Zrnic & Candes (2024), which explicitly uses the current features $x_t$ to determine the query probability.

ACKNOWLEDGEMENTS

The authors thank the anonymous reviewers for their constructive suggestions, which helped enrich the discussion of related work and clarify the experimental setup. The authors also appreciate the support by NSF CCF-2403392, as well as by the Google Gemma Academic Program and Google Cloud Credits.

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

## A  Experiments on the Effect of the Mixing Parameter

In this section, we report the results of experiments on tuning the mixing constant in the mixture policy of Zrnic & Candes (2024), evaluated across four different datasets. A detailed description of these datasets is provided in Section 6 and Appendix E.

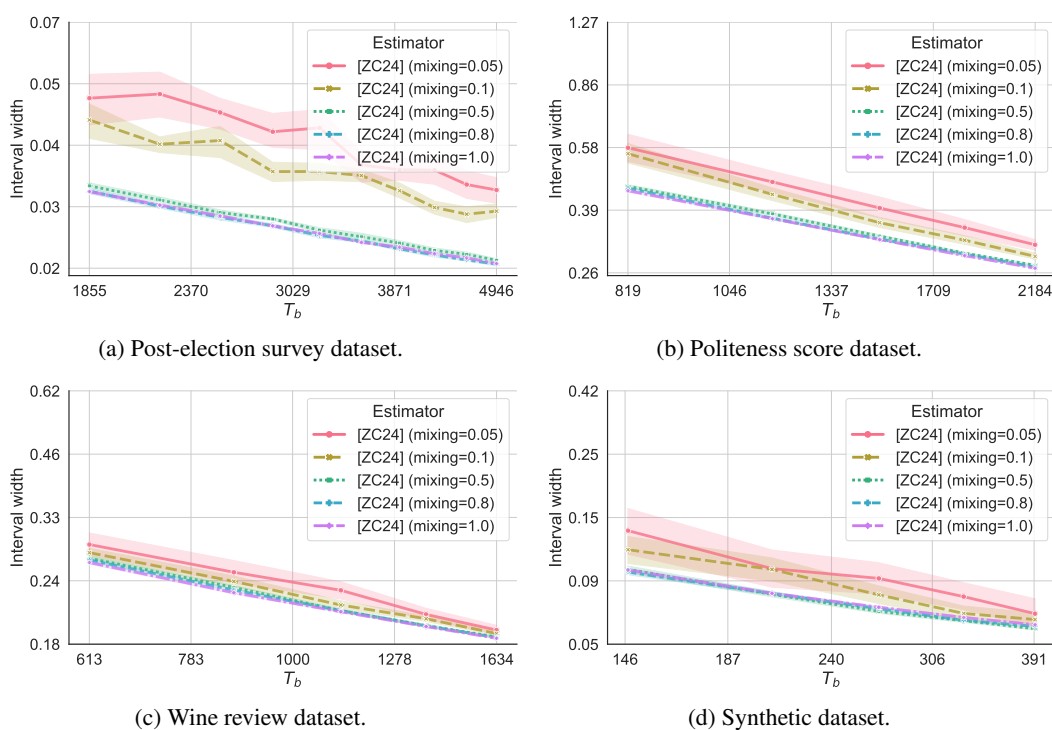

Figure 5: Interval width vs. the sampling budget parameter $T_b$ for different values of the mixing parameter of the query probability scheme in Zrnic & Candes (2024). Averaged over 50 repeated runs.

## B  Extended Related Work

**Active Statistical Inference.** The ideas in this work are motivated by the recent approach of Active Statistical Inference (Zrnic & Candes, 2024). Extending this framework, Angelopoulos et al. (2025) propose a method that optimizes the sampling rate between gold-standard and pseudo-labels rather than relying on a fixed label budget, and derive an improved active sampling policy. A recent work by Gligorić et al. (2025) utilizes LLM verbalized confidence scores to guide their sampling policy and subsequently performs active inference by combining the LLM and human annotations.

**Prediction Powered Inference.** The Active Statistical Inference framework is grounded in the idea of Prediction Powered Inference (PPI) Angelopoulos et al. (2023a), which differs from the former in that it assumes the availability of a small, pre-labeled dataset. The work of Angelopoulos et al. (2023b) introduces PPI++, which improves the computational efficiency of PPI by addressing the intractability of the original confidence interval construction and using a tuning parameter to control the influence of the model predictions based on their quality. Subsequent works (Dorner et al., 2025; Mani et al., 2025) analyze the critical role of the correlation between the gold-standard and model-generated labels for the performance of PPI. Focusing on the few-label regime, Eyre & Madras (2025) argue that the PPI++ framework may lead to a significantly biased estimator that is less efficient than classical inference by establishing its connection to univariate ordinary least squares regression. Datta & Polson (2025) examine the use of an inverse probability weighted (IPW) bias-correction term in the PPI mean estimator, inspired by classical Horvitz–Thompson and Hájek estimators. The work of De Bartolomeis et al. (2025) establishes a connection of PPI++ with the

augmented inverse probability weighting (AIPW) estimator and propose an extension, which allows utilizing predictions from multiple foundation models. To address cases where the model accuracy varies across subdomains, Fisch et al. (2024) apply a stratification approach to PPI. Zrnic (2024) explore a bootstrap-based PPI variation to tackle arbitrary estimation problems. Xu et al. (2025) improve the PPI framework by proposing a safe PPI estimator that is always more efficient than the initial supervised estimator and can be used for arbitrary inferential problems. Li et al. (2025) generalize the ideas of PPI to a dynamic performative setting and show improved confidence regions in the task of performative prediction. In a semi-supervised context, Zrnic & Candès (2023) propose a method of using the labeled datapoints for cross-fitting and using the fitted models to compute the desired estimator. Cortinovis & Caron (2025) extend PPI by applying a Bayes-assisted framework that uses prior knowledge on the accuracy of the model predictions. In the case of compound estimation settings, Li & Ignatiadis (2025) adopt an approach that combines PPI with empirical Bayes shrinkage to correct noisy predictions within each problem and subsequently uses these as a shrinkage target. An interesting work by Csillag et al. (2025) presents a PPI framework based on e-values. Other applications of the PPI framework include LLM-assisted rank-set construction (Chatzi et al., 2024), average treatment effects from multiple datasets (Wang et al., 2025), clinic trial outcomes (Poulet et al., 2025), evaluating the accuracy of machine learning systems (Boyeau et al., 2025; Park et al., 2025), machine learning generated surrogate rewards for multi-armed bandits (Ji et al., 2025b). A few other works have explored alternative machine-learning assisted estimators, e.g. Schmutz et al. (2023); Egami et al. (2023); Miao et al. (2023); Miao & Lu (2024); Gan et al. (2024).

**Active Learning.** Similar to the Active Statistical Inference protocol is active learning (Settles, 2009; Dasgupta, 2011; Hanneke, 2014), which frames learning as a process in which a machine learning model selectively queries unlabeled instances to be labeled by an oracle. In contrast to Active Statistical Inference, which focuses on enhancing statistical inference, the goal of active learning is to improve the predictive power of the machine learning model through the strategic use of labeled data. One of the main approaches in active learning, and the one most closely aligned with active estimation, is the uncertainty sampling strategy (Schohn & Cohn, 2000; Tong & Koller, 2000; Tur et al., 2005; Joshi et al., 2009; Gal et al., 2017; Ducoffe & Precioso, 2018; Beluch et al., 2018; Ren et al., 2021), where the model aims to query the labels of the most informative data points, i.e., the ones which the model is most uncertain about.

**Importance Sampling.** The idea of prioritizing the most influencial samples is also closely related to adaptive importance sampling (Owen, 2013), which is a sequential scheme that updates the proposal distribution by learning from previously sampled values in order to better approximate some property of a target distribution. Active mean estimation is analogous to adaptive importance sampling, in that it adaptively selects samples that contribute most to reducing the variance of the estimator based on previously observed data to improve the accuracy of the mean estimator.

**Non-Asymptotic Results in Zrnic & Candes (2024).** We acknowledge that in Appendix C in Zrnic & Candes (2024), the authors consider incorporating the notion of actively querying the ground truth into the technique for estimating means of bounded random variables proposed by Waudby-Smith & Ramdas (2024). Waudby-Smith & Ramdas (2024) leverage the duality between sequential hypothesis testing and the construction of confidence intervals. More concretely, their method reduces the task of constructing a confidence interval for the mean to a potentially infinite number of hypothesis testing problems. Each hypothesis testing problem corresponds to whether the observed samples have a population mean equal to a specific value. Hence, for a continuous random variable, this corresponds to an infinite number of hypotheses. In practice, a discretization is used. The confidence interval $\mathcal{C}_t$ at time $t$ then consists of those hypothesized mean values that have not been rejected based on the data observed up to time $t$, i.e., $\mathcal{C}_t := \{\nu : H_0^{(\nu)} \text{ is not rejected based on observations up to time } t\}$, where $H_0^{(\nu)}$ denotes the hypothesis that the population mean is $\nu$. While Zrnic & Candes (2024) provide the valuable idea of integrating these techniques and also provide some simulation results, the specific step-by-step algorithmic details and theoretical guarantees for active sequential mean estimation remain to be explicitly elaborated.

## C  PROOFS OF THE THEORETICAL RESULTS IN SECTION 4

**Theorem 1.** *Fix a time horizon $T \geq 4$. Assume each random variable $g_t$ is bounded, i.e., $|g_t| \leq G$ for a constant $G > 0$. Denote $\sigma_t^2 := \mathbb{E}\left[(g_t - \mu_y)^2 | \mathcal{F}_{t-1}\right]$ the conditional variance, where $\mathcal{F}_{t-1}$ is the filtration up to $t - 1$. Then, for any $\delta \in (0, 1/e)$, with probability at least $1 - \delta$, $\forall t \in [T]$ :*

$$|w_{t+1} - \mu_y| \leq \frac{2\max\left\{2\sqrt{S_t}, (G + |\mu_y|)\sqrt{\log\left(\frac{\log(T)}{\delta}\right)}\right\}\sqrt{\log\left(\frac{\log(T)}{\delta}\right)}}{T} + \left(1 - \frac{t}{T}\right)|\mu_y|,$$

*where $S_t = \sum_{s=1}^{t} \sigma_s^2$.*

*Proof.* The difference of $w_t - \mu_y$ can be decomposed into two terms. Specifically, we have

$$w_{t+1} - \mu_y = \left(\frac{1}{T}\sum_{s=1}^{t} g_s\right) - \mu_y = \frac{1}{T}\sum_{s=1}^{t}(g_s - \mu_y) - \left(1 - \frac{t}{T}\right)\mu_y. \tag{7}$$

Let us analyze the second-to-last-term $\sum_{s=1}^{t}(g_s - \mu_y)$ on (7). We note that $(g_s - \mu_y)_{s \geq 1}$ forms a martingale difference sequence, i.e., $\mathbb{E}[g_s - \mu_y | \mathcal{F}_{s-1}] = 0$, where $\mathcal{F}_{s-1}$ encodes all information up to time $s - 1$. Furthermore, $\forall s : |g_s - \mu_y| \leq G + |\mu_y|$. Also, the conditional variance is

$$\text{Var}(g_s - \mu_y | \mathcal{F}_{s-1}) = \mathbb{E}\left[(g_s - \mu_y)^2 | \mathcal{F}_{s-1}\right] = \sigma_s^2$$

By Freedman's inequality (Lemma 1), we have, with probability $1 - \delta$,

$$\forall t \in [T] : \sum_{s=1}^{t}(g_s - \mu_y) \leq 2\max\left\{2\sqrt{\sum_{s=1}^{t}\sigma_s^2}, (G + |\mu_y|)\sqrt{\log(\log(T)/\delta)}\right\}\sqrt{\log(\log(T)/\delta)}. \tag{8}$$

Combining (7) and (8), we obtain the following holds simultaneously at all $t \in [T]$, with probability at least $1 - \delta$:

$$|w_{t+1} - \mu_y|$$
$$\leq \frac{2\max\left\{2\sqrt{\sum_{s=1}^{t}\sigma_s^2}, (G + |\mu_y|)\sqrt{\log(\log(T)/\delta)}\right\}\sqrt{\log(\log(T)/\delta)}}{T} + \left(1 - \frac{t}{T}\right)|\mu_y|.$$

$\square$

## D  PROOFS OF THE THEORETICAL RESULTS IN SECTION 5

**Lemma 2.** *The conditional variance has the following decomposition:*

$$\mathbb{E}\left[(g_t - \mu_y)^2 | \mathcal{F}_{t-1}\right]$$
$$= \mathbb{E}\left[f_t(x_t)^2 \Big| \mathcal{F}_{t-1}\right] + \mathbb{E}\left[(y_t - f_t(x_t))^2 \frac{1}{\pi_t(x_t)} \Big| \mathcal{F}_{t-1}\right] + 2\mathbb{E}\left[f_t(x_t)(y_t - f_t(x_t)) \Big| \mathcal{F}_{t-1}\right] - \mu_y^2.$$

*Furthermore, assume that the query policy at time $t$ is $\mathcal{F}_{t-1}$-measurable, i.e., there exists a random variable $p_t \in [0, 1]$, measurable with respect to $\mathcal{F}_{t-1}$, such that $\pi_t(x_t) = p_t$. Then, we have*

$$\mathbb{E}\left[(y_t - f_t(x_t))^2 \frac{1}{\pi_t(x_t)} \Big| \mathcal{F}_{t-1}\right] = \frac{1}{p_t}\mathbb{E}\left[(y_t - f_t(x_t))^2 \Big| \mathcal{F}_{t-1}\right].$$

*Proof.*

$$\mathbb{E}\left[(g_t - \mu_y)^2 | \mathcal{F}_{t-1}\right]$$

$$= \mathbb{E}\left[g_t^2 | \mathcal{F}_{t-1}\right] - \mu_y^2$$

$$= \mathbb{E}\left[f_t(x_t)^2 \middle| \mathcal{F}_{t-1}\right] + \mathbb{E}\left[(y_t - f_t(x_t))^2 \frac{\xi_t^2}{\pi_t^2(x_t)} \middle| \mathcal{F}_{t-1}\right] + 2\mathbb{E}\left[f_t(x_t)(y_t - f_t(x_t))\frac{\xi_t}{\pi_t(x_t)} \middle| \mathcal{F}_{t-1}\right]$$

$$- \mu_y^2$$

$$= \mathbb{E}\left[f_t(x_t)^2 \middle| \mathcal{F}_{t-1}\right] + \mathbb{E}\left[(y_t - f_t(x_t))^2 \frac{1}{\pi_t(x_t)} \middle| \mathcal{F}_{t-1}\right] + 2\mathbb{E}\left[f_t(x_t)(y_t - f_t(x_t)) \middle| \mathcal{F}_{t-1}\right] - \mu_y^2,$$

where the first equality follows from that $\mathbb{E}\left[g_t | \mathcal{F}_{t-1}\right] = \mu_y$. $\qquad\square$

**Lemma 5.** *Assume that the range of oracle's output is bounded, i.e., $\forall t : \Phi(x_t) \leq B$, for a constant $B > 0$. Then, $\forall t : \left|\dot{\ell}_t\right|^2 \leq \frac{B^2}{\beta^4}$.*

*Proof.*

$$\forall t : \quad \left|\dot{\ell}_t\right|^2 = \frac{\Phi_t^2(x_t)}{p_t^4} \leq \frac{B^2}{p_t^4} \leq \frac{B^2}{\beta^4},$$

where the first inequality follows from that $\forall t : \Phi(x_t) \leq B$, and the last inequality uses that $\forall t : p_t \geq \beta$. $\qquad\square$

**Theorem 2.** *Assume that there is an oracle that outputs $\Phi_t(x_t)$ at each $t$ such that $\frac{1}{c_1}\Phi_t(x_t) \leq \mathbb{E}\left[(y_t - f_t(x_t))^2 \middle| \mathcal{F}_{t-1}\right] \leq c_0 \Phi_t(x_t)$ for some constant $c_0, c_1 > 0$ and that $\forall t : \Phi(x_t) \leq B$. Set the parameter $\gamma = \frac{1}{\sqrt{T}}\frac{\beta^2}{B}$. Using the query policy (4), we have that, with probability at least $1 - \delta$, $\forall t \in [T]$:*

$$|w_{t+1} - \mu_y| \leq \frac{2\max\left\{2\sqrt{\Psi_t}, (G + |\mu_y|)\sqrt{\log\left(\frac{\log(T)}{\delta}\right)}\right\}\sqrt{\log\left(\frac{\log(T)}{\delta}\right)}}{T} + \left(1 - \frac{t}{T}\right)|\mu_y|,$$

*for any $\beta \in (0, \tau)$ and $\tau \in (0, 1)$, where $\Psi_t \leq c_0 c_1 \sigma_{1:t}^{*2} + 2c_0 \frac{t}{\sqrt{T}}\frac{B}{\beta^2} + \frac{c_0(R(p_{1:t}^*) - \min_{p \in \mathcal{K}} R(p))\sqrt{T}B}{\beta^2}$.*

*Proof.* By Lemma 2 and the constraint that the query probability $p_t$ is fully determined in $\mathcal{F}_{t-1}$, we have

$$\sigma_t^2 = \mathbb{E}\left[f_t(x_t)^2 \middle| \mathcal{F}_{t-1}\right] + \frac{1}{p_t}\mathbb{E}\left[(y_t - f_t(x_t))^2 \middle| \mathcal{F}_{t-1}\right] + 2\mathbb{E}\left[f_t(x_t)(y_t - f_t(x_t)) \middle| \mathcal{F}_{t-1}\right] - \mu_y^2$$

$$\leq \mathbb{E}\left[f_t(x_t)^2 \middle| \mathcal{F}_{t-1}\right] + c_0\frac{\Phi_t(x_t)}{p_t} + 2\mathbb{E}\left[f_t(x_t)(y_t - f_t(x_t)) \middle| \mathcal{F}_{t-1}\right] - \mu_y^2$$

$$= \mathbb{E}\left[f_t(x_t)^2 \middle| \mathcal{F}_{t-1}\right] + c_0\left(\frac{\Phi_t(x_t)}{p_{1:t}^*} + \frac{\Phi_t(x_t)}{p_t} - \frac{\Phi_t(x_t)}{p_{1:t}^*}\right) + 2\mathbb{E}\left[f_t(x_t)(y_t - f_t(x_t)) \middle| \mathcal{F}_{t-1}\right]$$

$$- \mu_y^2$$

$$\leq \mathbb{E}\left[f_t(x_t)^2 \middle| \mathcal{F}_{t-1}\right] + \frac{c_0 c_1}{p_{1:t}^*}\mathbb{E}\left[(y_t - f_t(x_t))^2 \middle| \mathcal{F}_{t-1}\right] + c_0\left(\frac{\Phi_t(x_t)}{p_t} - \frac{\Phi_t(x_t)}{p_{1:t}^*}\right)$$

$$+ 2\mathbb{E}\left[f_t(x_t)(y_t - f_t(x_t)) \middle| \mathcal{F}_{t-1}\right] - \mu_y^2$$

$$\leq c_0 c_1 \sigma_{t,(t)}^{*2} + c_0\left(\frac{\Phi_t(x_t)}{p_t} - \frac{\Phi_t(x_t)}{p_{1:t}^*}\right),$$

where the last inequality uses the fact that $c_0 c_1 \geq 1$. We note that the above inequality holds for all $t$. Hence, we have

$$\sum_{s=1}^{t} \sigma_s^2 \leq c_0 c_1 \sigma_{1:t}^{*2} + c_0 \text{Regret}_t(p_{1:t}^*), \tag{9}$$

by summing up the above inequality for each round. To proceed, we use the regret bound that we have from Lemma 4:

$$
\begin{aligned}
\text{Regret}_t(p_{1:t}^*) &\leq 2\gamma \sum_{s=1}^{t} \left| \dot{\hat{\ell}}_s \right|^2 + \frac{R(p_{1:t}^*) - \min_{p \in \mathcal{K}} R(p)}{\gamma} \\
&\overset{(i)}{\leq} 2\gamma t \frac{B^2}{\beta^4} + \frac{R(p_{1:t}^*) - \min_{p \in \mathcal{K}} R(p)}{\gamma} \\
&\overset{(ii)}{=} 2\frac{t}{\sqrt{T}} \frac{B}{\beta^2} + \frac{(R(p_{1:t}^*) - \min_{p \in \mathcal{K}} R(p)) \sqrt{T} B}{\beta^2},
\end{aligned} \tag{10}
$$

where (i) is from Lemma 5 and (ii) is by the choice of $\gamma = \frac{1}{\sqrt{T}} \frac{\beta^2}{B}$. Combining (9) and (10), we have

$$\sum_{s=1}^{t} \sigma_s^2 \leq c_0 c_1 \sigma_{1:t}^{*2} + 2c_0 \frac{t}{\sqrt{T}} \frac{B}{\beta^2} + \frac{c_0 \left( R(p_{1:t}^*) - \min_{p \in \mathcal{K}} R(p) \right) \sqrt{T} B}{\beta^2}.$$

Using the above bound together with Theorem 1 leads to the result. This completes the proof.

$\square$

# E    ADDITIONAL EXPERIMENTAL DETAILS

## E.1    SYNTHETIC DATASET

The fourth dataset used in the experiments is a synthetic dataset that is generated for binary classification according to a logistic model. More specifically, the covariates $x_t \in \mathbb{R}^d$ are independently drawn from a multivariate normal distribution with zero mean and identity covariance $I_d$, where $d = 10$. The true parameter vector $w^*$ is sampled independently from a normal distribution with zero mean and covariance $0.5 \cdot I_d$. Gaussian noise $\epsilon_t \sim \mathcal{N}(0, 10^{-5})$ is added to each $x_t^\top w^*$ to produce the logits. The correspondig binary labels $y_t \in \{0, 1\}$ are then generated according to $y_t \sim \text{Bernoulli}\left( \sigma \left( x_t^\top w^* + \epsilon_t \right) \right)$, where $\sigma(z) = 1/(1 + e^{-z})$ denotes the sigmoid function. The ML model $f_t(\cdot)$ is implemented as a logistic regression model whose uncertainty predictor is estimated in the same way as in the post-election survey dataset, i.e., $u_t(x_t, f_t(\cdot)) = 2 \min\{f_t(x_t), 1 - f_t(x_t)\}$ using the predicted probabilities of $f_t(\cdot)$. A linear regression model is trained to predict the approximation oracle $\Phi(x_t)$ as in the previous tasks.

Figure 6 shows the experimental results on the synthetic dataset.

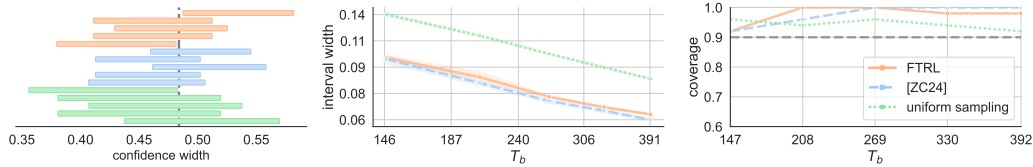

Figure 6: **Synthetic dataset**. Left: Intervals of randomly selected trials. Middle: Average confidence width across repeated trials vs. sampling budget $T_b$. Right: Percentage of trials that cover the true mean.

## E.2    EXPERIMENTAL SETUP

All experiments were repeated over 50 trials, and reported results correspond to the averages across these trials. At the start of each trial, the data points were randomly permuted. For each experiment,

the budget $T_b$ was varied over five uniformly spaced values between 15% and 40% of the total number of data points $T$. The interval width and coverage plots were obtained by linearly interpolating between the values at these grid points.

In the experimental setup, the ML model $f_t(\cdot)$, uncertainty estimator $u_t(\cdot, \cdot)$, and oracle $\Phi_t(\cdot)$ were updated periodically after observing a batch of $B$ data points. For the first two datasets (politeness score and wine review analysis), the estimators were updated $N = 50$ times, while for the last two datasets (post-election survey and synthetic data), they were updated $N = 10$ times. Accordingly, the batch size was set to $B = \text{round}\left(\frac{T_b}{N}\right)$.

For the FTRL plicy, we set the upper bound hyperparameter to $\tau = \frac{T_b}{T}$ in accordance with our theoretical analysis to ensure that the query probability satisfies the sampling constraint. The lower bound hyperparameter was set to $\beta = \frac{\tau}{8} > 0$, to prevent the sampling probability from becoming too small, and thereby encouraging exploration, while still remaining sufficiently below $\tau$ so that the resulting sampling interval is non-trivial and allows the algorithm to adjust the sampling probability over time. Furthermore, the hyperparameter $\gamma$ was chosen as $\gamma = \frac{1}{\sqrt{T}}$, in line with common practice in online learning, to guarantee sublinear regret growth with respect to $T$, which is necessary for achieving no-regret performance. For the policy of Zrnic & Candes (2024), we set the mixing hyperparameter to $\lambda = 0.5$, which is the recommended value used in their experimental setup, to enable a comparison with their proposed policy.

