# OpenReview forum: "Revisiting Active Sequential Prediction-Powered Mean Estimation"
_ICLR.cc/2026/Conference — ICLR 2026 Poster_

### Official Review · Reviewer_Zfwx · 2025-10-31

**Soundness:** 3
**Presentation:** 3
**Contribution:** 2
**Rating:** 6
**Confidence:** 2

**Summary:**

This paper revisits the problem of active sequential prediction-powered mean estimation, where a learner estimates the mean of an unknown label distribution under a limited labeling budget by adaptively deciding whether to query true labels or rely on model predictions. Building on prior work (Zrnic & Candès, 2024), the authors make an intriguing empirical observation that confidence intervals are often narrowest when the query policy is nearly uniform, ignoring model uncertainty. Motivated by this, they develop a non-asymptotic analysis providing time-uniform, data-dependent confidence bounds for the estimator and reformulate the label-querying process as an online convex optimization problem solved via the Follow-the-Regularized-Leader (FTRL) algorithm. Their theoretical results show that the FTRL policy converges to the maximal allowable query rate and that a uniform querying strategy is asymptotically optimal. Experiments across real and synthetic datasets confirm that this simple strategy matches or outperforms uncertainty-based methods in confidence width and coverage, revealing that uncertainty-aware sampling may offer limited benefits for sequential mean estimation.

**Strengths:**

The paper is original in revisiting active sequential mean estimation through a new theoretical lens, uncovering an unexpected empirical pattern that challenges prior assumptions about the value of model uncertainty. Its non-asymptotic analysis fills a key theoretical gap left by earlier asymptotic-only results, providing rigorous, time-uniform, and data-dependent confidence bounds that enhance understanding of estimator behavior. The introduction of a no-regret online learning formulation using FTRL to govern the query policy is conceptually elegant and technically sound, connecting active inference with online convex optimization theory. The experimental validation is thorough and consistent with the theoretical findings, strengthening the overall credibility of the claims. The paper is also clearly written and well-organized, effectively bridging empirical observation, theoretical reasoning, and practical implications—making it a significant and high-quality contribution to the fields of active statistical inference and prediction-powered estimation.

**Weaknesses:**

While the paper is theoretically well-grounded, several aspects could be improved to strengthen its impact. First, the novelty is somewhat incremental, as the core setup largely extends the framework of Zrnic & Candès (2024) by introducing a non-asymptotic analysis and reformulating the query policy using standard online learning tools (FTRL). The contribution could be made more compelling by deeper theoretical insight into why uncertainty-based sampling fails in practice, beyond the observed convergence of the no-regret policy. Second, the experimental scope is limited, the datasets used (text politeness, wine review, post-election survey, and one synthetic set) are small-scale and do not test generalization to high-dimensional or noisy real-world domains where uncertainty estimation might matter more. Adding experiments in such settings or sensitivity analyses for different noise structures would better validate the theoretical claims. Third, the discussion of assumptions and limitations is brief; for example, the oracle approximation and boundedness assumptions may not hold in practical scenarios, and their empirical impact is not fully analyzed. Finally, connections to related frameworks such as active learning or adaptive importance sampling[1][2] could be discussed more thoroughly to position the work within the broader literature. Addressing these points would make the contribution more robust and its insights more broadly applicable.

[1] Pareto smoothed importance sampling, jmlr 2025

[2] Active Advantage-Aligned Online Reinforcement Learning with Offline Data, arxiv 2025

**Questions:**

see weakness

---

> ### Author Response · Authors · 2025-11-20
>
> We thank the reviewer for their constructive feedback. Our responses to the points raised are provided below.
>
> - **Theoretical insights**: We would like to respectfully highlight that while the problem setup is the same as that in [1], our contributions are motivated by the empirical performance of the query policy proposed in [1] (see, e.g., Figure 1 in our submission). Consequently, we first conduct a non-asymptotic analysis of their estimator, which in turn inspires us to propose a new and different query policy with theoretical guarantees (Theorem 2 in the paper). Our proposed policy is to use FTRL, which naturally incorporates the past covariates and past uncertainty estimates and admits a simple closed-form update (Lemma 3 in our paper). In contrast, the sampling rule in [1] chooses the sampling probability at each step using an estimate of the model’s uncertainty at the current covariate.
>
> - **Datasets**: The motivation of our work is based on an empirical observation regarding  the query-policy scheme proposed in the prior work (Figure 1). Although the feature dimension of the datasets such as the one also used in the prior work (Figure 1) is of only modest size, the intriguing pattern already exists. However, we agree that testing whether this intriguing pattern also persists for datasets with high-dimensional feature size can be a valuable direction.
>
> - **Discussion of assumptions and limitations**: We want to emphasize that our assumptions are sufficiently general and capture all cases of interest. In particular, the oracle approximation assumption:
> $$
> \frac{1}{c_1} \Phi_t(x_t) \leq \mathbb{E}\left[ \left( y_t - f_t(x_t) \right)^2 \Bigg| \mathcal{F}_{t-1} \right] \leq c_0 \Phi_t(x_t)
> $$
> relies on constants $c_0$ and $c_1$. Our theoretical result in Theorem 2 accordingly incorporates these constants into the final bound, allowing it to adapt to the quality of the oracle’s variance approximation. The same is true for the boundedness assumption $\Phi_t(x_t) \leq B$. While the efficiency of the active mean estimation framework relies on the quality of the oracle approximation, we note that the oracle $\Phi_t(x_t)$ generally becomes more accurate as more data are collected. In the experimental setup, following our protocol, we initially use a small portion of the data to train an approximation model for this purpose. Notably, the effectiveness of the approximator depends on the variability and representativeness of the data used during training.
>
> - **Connection to active learning and adaptive importance sampling**:  We thank the reviewer for this suggestion that helps improve our work. We have updated Appendix B in the revised paper accordingly with a discussion of relevant related work in these areas.
> We can also refer the reader to the work of Zrnic and Candès.\
> In their section on related works, the authors have addressed the connection between the problem setting that our papers concern and active learning as well as adaptive importance sampling.
>
> [1] Zrnic, T. and Candès, E., Active Statistical Inference

---

### Official Review · Reviewer_nFbY · 2025-11-01

**Soundness:** 4
**Presentation:** 3
**Contribution:** 4
**Rating:** 8
**Confidence:** 3

**Summary:**

The paper revisits the problem of active sequential prediction-powered mean estimation — estimating the mean of a label $y$
$y$ when each observation’s label may or may not be queried. If the label is not queried (to save cost), a machine-learning model’s prediction is used instead and there is finite budget $T_b \ll T$ of queries allowed. Here are the follewing finding.

1) Based on Freedman's theorem (a version of Bernstein concentration inequality) the author provide non-asymptotic confidence intervals
and data-dependent confidence bound valid at any step.

2) Reinterpret query-probability selection as a no-regret online learning problem (via the Follow-the-Regularized-Leader algorithm).

3) The discovery  that under sequential settings, uniform querying (without conditioning on covariates) may be both theoretically justified and empirically optimal.

**Strengths:**

1. Rigorous theoretical analysis based on martingale difference concentration inequality.

2. New connection with no-regret online learning problem which suggest practical algorithm

3. Clear empirical insights that uniform querying performs comparably (or better) than uncertainty-based querying which may simulate further work.  Since this optimal strategy is also very simple and easy to implement this is a significant finding.

4. Very well written paper combing theory, algorithmic development and experimental findings.

Overall the paper looks very strong to the reviewer (who is not an expert in this field, especially online learning).

**Weaknesses:**

No obvious one.

**Questions:**

Have you considered extending your framework to other risk functionals (e.g., variance or quantile estimation) beyond the mean, and would the same uniform-querying phenomenon persist?

---

> ### Author Response · Authors · 2025-11-20
>
> We thank the reviewer for the nice summary of our work and the positive feedback.
>
>
> Regarding the reviewer's insightful question, we have indeed begun thinking about extending our analysis to other risk functionals, as the reviewer mentioned.
>
> More concretely, for other loss functions,
> we think the first step might be to obtain a non-asymptotic analysis
> of the following point estimate proposed in the prior work of Zrnic and Candes:
> $$
> w_t = \\arg\\min_w \sum_{s=1}^{t-1}  \tilde{\ell}_s(w),
> \text{ where }
>  \tilde{\ell}_s(w) = \ell(w; x_s, f_s(x_s)) + \frac{\ell(w; x_s, y_s) - \ell(w; x_s, f_s(x_s))}{\pi_s(x_s)}  \xi_s,
> $$
> where $\xi_s \sim \text{Bernoulli}(\pi_s(x_s))$, $(x_s, y_s)$ is the feature-label pair at time $s$, $f_s(x_s)$ is the model prediction on $x_s$, and $\ell(w; x_s, y_s)$ denotes the loss function  evaluated at $w$ for the feature-label pair $(x_s, y_s)$.
> From there, we may be able to gain further insights into how the no-regret learning approach can be leveraged to design query probabilities with some theoretical guarantees.
>
> However, we feel that a new set of tools may be necessary to make progress in this direction. In particular, we think that there are some technical challenges. For example, if the model $f_t(\cdot)$
>  used to label the unlabeled points
> or uncertainty predictor $\pi_t(\cdot)$
>  is updated by using previously collected data $(x_s, y_s)$ for $s < t$, then the components of the loss functions in the finite sum $\sum_{s=1}^{t-1} \tilde{\ell}_s(w)$ become dependent. This dependency may prevent a straightforward adaptation of results on empirical risk minimization from learning theory to the setting of active statistical inference beyond mean estimation.
> We also think that whether the same uniform-querying phenomenon persists across other risk functionals requires comprehensive simulations to answer, which we plan to investigate next.

---

### Official Review · Reviewer_4Zjz · 2025-11-02

**Soundness:** 3
**Presentation:** 2
**Contribution:** 3
**Rating:** 4
**Confidence:** 3

**Summary:**

This paper revisits the active statistical inference methods proposed by Zrnic & Candes (2024)--where that previous paper presents methods for prediction powered inference (Angelopoulos et al 2023a) under active data selection--and the current paper claims the following contributions: (1) Empirically exploring the effect of different values of the mixture weight (ie, a mixture weight on the active data selection policy) in Zrnic & Candes (2024) and finding that the sharpest intervals tended to occur with more weight on the constant probability; (2) they present non-asymptotic analysis of the mean estimator with a data-dependent bound; (3) analysis on convergence when a no-regret learning approach is used; (4) simulation experiments to support the theoretical results.

**Strengths:**

This paper studies an important problem of how to do valid statistical inference under active data collection, building on prior work of active statistical inference (Zrnic & Candes 2024). The observations about the role of the mixture parameter, the non-asymptotic analysis, and the analysis on convergence seem like they would be of interest to the community. The writing is overall clear and the analysis seems sound. Overall I enjoyed reading the paper.

**Weaknesses:**

**Adding acknowledgement & discussion of prior non-asymptotic version of Zrnic & Candes (2024):** While the paper does have an overall fairly thorough discussion of related work and its relationship to the prior work of Zrnic & Candes (2024), there is one key aspect of its discussion that seems like it needs to be updated, to acknowledge that Zrnic & Candes (2024) do have some consideration of non-asymptotic versions of their method, which is the main claimed contribution of the current paper. That is, at the top of Page 2 the current paper states “The non-asymptotic analysis of the estimator was missing in the prior work [Zrnic & Candes (2024)]”; however, Appendix C in Zrnic & Candes (2024) is titled “Non-asymptotic results” and it both explains how its main results could extend directly to non-asymptotic analogs, as well as presents some experimental results of non-asymptotic variants. I understand that Zrnic & Candes (2024) do not comprehensively or explicitly write out the formal analysis for the non-asymptotic variants, but given that this is the core claimed contribution of the current paper, I think that the current paper needs to be revised to acknowledge this prior consideration.

**Questions:**

Can the authors explain how their non-asymptotic analysis and results relate to Appendix C in Zrnic & Candes (2024)? I think that such discussion should also be added to the paper.

---

> ### Author Response · Authors · 2025-11-20
>
> We are thankful to hear that the reviewer enjoys reading our paper.
>
>
> Regarding the addition of an acknowledgement and discussion about Appendix C (Section Title: “Non-Asymptotic Results”) in Zrnic \& Candès (2024), we agree with the reviewer’s excellent suggestion and have incorporated this material into Appendix B of our revised paper.
>
> Furthermore, we have updated the sentence in the introduction to soften the tone as:\
> *"The non-asymptotic analysis of the estimator was not provided in the prior work."*
>
> We also have added a footnote on the same page:\
> *"However, we acknowledge that Appendix C of Zrnic \& Candès (2024) outlines a related scheme based on estimating bounded means via testing by betting (Waudby \& Ramdas), which may come with certain non-asymptotic guarantees. We provide further discussion in Appendix B."*
>
> On the other hand, respectfully, we believe that the method in Section 6 of Zrnic \& Candès (2024) and the method outlined in Appendix C of the same paper are different. We stated that the non-asymptotic analysis of their proposed estimator (that appears in their main text) was not provided, where the update of the estimator is also replicated in (1) of our paper. In particular, a concrete bound on $|w_t - \mu_y|$ under a query policy $\pi(x_t)$ at any time $t$ was not provided; they only provided an asymptotic analysis in their Proposition~6.1.
>
>
> For Appendix C in Zrnic \& Candès (2024), the idea they suggest is to incorporate the notion of actively querying the ground truth into the technique for estimating means of bounded random variables proposed by Waudby-Smith and Ramdas. Waudby-Smith and Ramdas leverage the duality between sequential hypothesis testing and the construction of confidence intervals. More concretely, their method reduces the task of constructing a confidence interval for the mean to a potentially infinite number of hypothesis testing problems. Each hypothesis testing problem corresponds to whether the observed samples have a population mean equal to a specific value. Hence, for a continuous random variable, this corresponds to an infinite number of hypotheses. In practice, a discretization is used. The confidence interval $C_t$ at time $t$ then consists of those hypothesized mean values that have not been rejected based on the data observed up to time $t$, i.e.,
> $$
> C_t := \\{ \nu : H_{0}^{(\nu)} \text{ is not rejected based on observations up to time } t \\},
> $$
> where $H_{0}^{(\nu)}$ denotes the hypothesis that the population mean is $\nu$.
> $\{ \}$
>
> As the reviewer may have noted or have already kindly pointed out,
> while Appendix C of Zrnic \& Candès (2024) introduces the nice idea, the step-by-step algorithmic description of how the betting-based technique is combined with active querying, along with detailed implementation specifications, remain to be elaborated.
> In particular, we note that Waudby-Smith and Ramdas's theoretical result is for a bounded $[0,1]$ random variable, but the random variable
> $f_t(X_t) + (Y_t - f_t(X_t)) \frac{\xi_t}{\pi_t(X_t)}$
> in their Appendix C can potentially fall outside this range in practical scenarios. Hence, Theorem 3 in Waudby-Smith and Ramdas needs to be modified to accommodate the scenario of active statistical inference.
> We believe that the confidence interval of the method suggested in their Appendix~C should depend on the upper bound $G$ of the magnitude of the random variable
> $f_t(X_t) + (Y_t - f_t(X_t)) \frac{\xi_t}{\pi_t(X_t)}.$
> Specifically, we believe the resulting confidence width at $t$ is $\tilde{O}(G/\sqrt{t})$, but we consider providing a theoretical guarantee for their method outlined in their Appendix C to be beyond the scope of our work.
>
>
> We view our non-asymptotic analysis as the first contribution of our work,
> which is motivated by the intriguing empirical patterns described in the preliminary section and also paves the way for the contributions presented in the later sections of our paper.
>
>
> We would appreciate it if the reviewer could let us know whether our response clarifies the difference, and if there are any lingering concerns.
> We would also greatly appreciate it if the reviewer could reconsider their evaluation accordingly after our acknowledgement and discussion based on the reviewer's suggestion in the updated paper.
>  Thank you!

---

### Official Review · Reviewer_2b77 · 2025-11-09

**Soundness:** 2
**Presentation:** 2
**Contribution:** 3
**Rating:** 4
**Confidence:** 2

**Summary:**

This paper revisits the active inference procedure proposed by Zrnic & Candes, 2024 [ZC24], where the goal is to better estimate the mean of a target variable with a query policy that is a mixing of uniform sampling and uncertainty-based sampling. This investigation is motivated by the empirical observation that uniform sampling yields tighter or comparable confidence intervals to uncertainty-based active sampling. A non-asymptotic upper bound on the L1 error of the active sequential prediction-powered mean estimation is provided, showing a rate of $O(1/\sqrt{t})$ at large sample sizes $t$. Then the paper demonstrates that a query policy independent of the current covariate and designed in a non-regret fashion converges to uniform sampling. This policy leads to similar confidence interval widths to the method proposed by Zrnic & Candes, 2024, in several real data experiments, where they both perform significantly better than uniform sampling.

**Strengths:**

- The analysis is motivated by an intriguing empirical observation that the active inference method [ZC24] performs comparably to uniform sampling in terms of the confidence interval width.

- It is an interesting conclusion that a query polity independent of the current covariate can produce similar confidence intervals to the active inference method [ZC24].

**Weaknesses:**

- There seem to be some contradictions in the empirical observations (See Questions).

- The motivation for looking at the non-regret approach is not sufficiently clear.

**Questions:**

- Does the method [ZC24] with mixing $\lambda=1$ correspond to the uniform sampling baseline in (6)? If so, why does the purple curve in Figure 1 differ significantly from the green one in Figure 4, while they both illustrate the results of uniform sampling on the post-election survey data set. Moreover, according to Figure 1, the method [ZC24] with a mixing policy of uniform and uncertainty-based samplings gives wider or comparable confidence intervals w.r.t. the uniform sampling baseline, which is the motivation of this article. However in Figures 2-4, the method [ZC24] yields consistently narrower confidence intervals than uniform sampling. Isn't there a contradiction?

- How are the hyperparameters $\gamma,\tau,\beta$ chosen for the non-regret method FTRL in the experiments of Section 6? Same question for the mixing parameter $\lambda$ for the method [ZC24].

---

> ### Author Response · Authors · 2025-11-20
>
> We thank the reviewer for the feedback and the important clarification questions.
>
> - **Motivation for FTRL approach**: The sequential active mean estimation approach of [1] is motivated by controlling the variance of the estimator. As shown in Lemma 2 of our paper, the only controllable term in the variance decomposition is $\mathbb{E} [ \left( y_t - f_t(x_t) \right)^2 \frac{1}{\pi_t(x_t)}  | \mathcal{F}_{t-1} ]$, which depends directly on the query probability. Since the problem is sequential in nature, controlling for this quantity can be achieved online by choosing a suitable query policy $\pi_t$ at each step, based on all the information gathered up to that point about the performance of the machine learning model. FTRL is a classical framework for making such sequential decisions, with the goal of selecting a query policy that performs well in hindsight based on all information available up to time $t-1$. In contrast to the sampling rule in [1], which chooses the sampling probability at each step using an estimate of the model’s uncertainty at the current covariate without observing its true label, our FTRL strategy incorporates the model’s performance on all previously sampled datapoints and their labels. Moreover, the FTRL formulation naturally integrates the budget constraint into the optimization process, avoiding the need for additional clipping or mixing with a uniform sampling rule.
>
> - **Empirical Observations**: We thank the reviewer for asking the clarification question.
> No, they are different updates; the curves represent different schemes. Specifically, we would like to clarify the distinction between the uniform sampling baseline in equation (6):
> $$w_T^{\text{Uniform}} = \frac{1}{T} \sum_{t=1}^T \left( f(x_t) + \frac{ (y_t - f(x_t)) \xi_t }{ T_b/T } \right), \text{where } \xi_t \sim \text{Bernoulli} \left( \frac{T_b}{T} \right)$$
> and the active mean estimator of Zrnic and Candes [1] with a uniform query policy (when $\lambda = 1$). For the uniform baseline (6), the method uses a fixed ML predictor $f(\cdot)$ provided at the beginning, as we described below (6) in the paper.
> In contrast, for the active mean estimator of [1], the ML predictor is updated periodically, i.e., the model $f_t(\cdot)$ is updated/retrained periodically, as illustrated in Algorithm 1 in our paper.  Zrnic \& Candes (2024) also have a similar comparison (where $\lambda=0.5$) to highlight the benefit of continuously collecting data and updating the ML predictor.
>
> (We note that there was a typo: the term $T_b/T$ was missing in the denominator of (6), which we have corrected in the updated version.)
>
> - **Hyperparameter Choice**: The upper bound hyperparameter $\tau$ was chosen as $\tau = \frac{T_b}{T}$ in accordance with our theory to ensure that the query probability satisfies the sampling constraint. The lower bound hyperparameter $\beta$ was chosen as $\beta = \frac{\tau}{8} > 0 $, to prevent the sampling probability from becoming too small, and thereby encouraging exploration, while still remaining sufficiently below $\tau$ so that the resulting sampling interval is non-trivial and allows the algorithm to adjust the sampling probability over time. Furthermore, the hyperparameter $\gamma$ was chosen as $\gamma = \frac{1}{\sqrt{T}}$, in line with common practice in online learning, to guarantee sublinear regret growth with respect to $T$, which is necessary for achieving no-regret performance. We set the hyperparameter $\lambda = 0.5$, which is the value recommended in [1] and adopted in their experimental setup, to enable a comparison between our algorithm and their update scheme. We have updated Appendix E accordingly to include these hyperparameter specifications.
>
> We hope our response sufficiently resolves the reviewer’s concerns about the theoretical and experimental aspects of the paper and provides a more accurate view of our work. We would appreciate it if the reviewer could reconsider their evaluation accordingly in light of these clarifications.
>
> [1] Zrnic, T. and Candès, E., Active Statistical Inference

---

> > ### Comment · Reviewer_2b77 · 2025-11-25
> >
> > I thank the authors for their response. Regarding the second point on the empirical observations, can I understand that Figure 1 actually shows that the benefit of the sequential active mean estimation over the uniform sampling baseline (6) observed in the paper of Zrnic & Candes, 2024 is induced by the updating of the prediction model $f$, not by the active sampling policy?

---

> > > ### Author Response · Authors · 2025-11-28
> > >
> > > It is not accurate to claim that the advantage of sequential active mean estimation over the uniform baseline arises solely from the updated prediction model. The policy of Zrnic and Candès (2024) still achieves comparable results by incorporating model uncertainty at the current covariate. As we discuss in our work, one potential contributing factor for the pattern in Figure 1 is that the estimated model variance can be large, making the uncertainty predictor less reliable in practice.
> > > Furthermore, as detailed in Section 2, the policy of Zrnic and Candès (2024) employs a few practical design choices that help satisfy the budget constraint without underutilizing it, while also preventing excessive estimator variance. These additions seem indeed to be effective in practice. However, integrating such rules into the policy introduces elements that are not directly derived from the underlying theoretical framework.
> > >
> > > Our theoretical and experimental results indicate that in this FTRL setting that we introduce, the optimal strategy based on all the information available up to $t-1$ is to simply set $p_t = T_b/T$. In particular, the uniform sampling policy $p_t = T_b/T$ emerges as the strategy required to achieve sublinear regret.
> > >
> > > Whether incorporating uncertainty estimates in some fundamentally different form of policy can lead to a significantly improved mean estimator is left for future investigation.

---

### Meta-Review · Area_Chair_aScf · 2026-01-05

**Summary:**

In this paper, the authors revisit the problem of actively estimating the mean of a label when there is a finite budget to query the labels of the observations. The work is motivated by the empirical observation that uniform sampling yields tighter or comparable confidence intervals to uncertainty-based active sampling. They then present non-asymptotic analysis of the mean estimator with a data-dependent bound. They also analyze the convergence when a no-regret learning approach is used. Finally, they provide simulation experiments to support their theoretical findings.

- The reviewers found the work well-written and theoretically well-grounded. They also appreciate that the analysis was motivated by an intriguing observation resulted from the an empirical study by the authors.

- The reviewers pointed out ways to improve the quality of the work, some of them were addressed by the authors during the rebuttal. I think they clarified the difference between their results and those by Zrnic and Candes (2024). I would suggest that the authors include this discussion in the paper and expand it to make it clear that their work is not simply an extension of Zrnic and Candes (2024). I also suggest that the authors expand the discussion on the assumptions and limitations of their theoretical framework, making it specifically clear which of the assumptions may not hold in practice. Finally, better describing the experimental results and more experiments to show how the work can be generalized to high-dimensions and to different noisy scenarios could further improve the quality of the paper.

**Reviewer Concerns:**

I think the authors addressed Reviewer 4Zjz's main concern regarding the difference between the results of the paper and the discussion on deriving non-asymptotic results in the appendix of Zrnic and Candes (2024). I also think the Reviewer 2b77's concerns were addressed, at least partially, by the authors.

**Reviewer Scores:**

Three reviewers did not react to the authors' response. Reviewer 2b77 asked further questions about one of the points they raised but did not change their score.

---

### Decision · Program_Chairs · 2026-01-26

Accept (Poster)